# The Role of Physical Activity, Enjoyment of Physical Activity, and School Performance in Learning Motivation among High School Students in Hungary

**DOI:** 10.3390/children9030320

**Published:** 2022-02-28

**Authors:** Tamás Berki, Zsófia Tarjányi

**Affiliations:** Institution of Physical Education and Sport Sciences, Faculty of Education, University of Szeged, 6725 Szeged, Hungary; zsofi2695@gmail.com

**Keywords:** physical activity, learning motivation, PACES, academic performance, enjoyment

## Abstract

The goal of this study was to analyze the role of physical activity, enjoyment of physical activity, and school performance in the dimensions of learning motivation (Cognitive Domain, Affective Domain, Moral Domain, Adult Pressure). A total of 249 high school students were involved in this study, ranging in age from 14 to 19 years old. A self-administered questionnaire was filled out by the participants, including questions about sociodemographic background and school performance. The International Physical Activity Questionnaire was used to assess physical activity. Learning motivation was measured by the School Motivation Inventory. The Physical Activity Enjoyment Scale was used to determine the enjoyment of physical activity. Path analyses were chosen as a statistical method to understand the associations between the variables. Our findings reveal that learning motivation was associated with school performance and enjoyment of physical activity. Physical activity did not show any direct relationship with learning motivation, but it was positively associated with school performance and enjoyment of physical activity, hence showing an indirect relationship with learning motivation. Another important finding was the role of physical activity enjoyment. It has a preventive role concerning pressure from adults since such enjoyment strengthens the intrinsic motivation of students. We believe our findings highlight the benefits of physical activity and enjoyment of physical activity in learning environments.

## 1. Introduction

It is well known that physical activity (PA) has a key role in preventing chronic disease and preserving mental health [1,2]. Several studies have been carried out in the past few years to provide empirical evidence of this phenomenon and have come to one conclusion: physical activity should start in childhood and adolescent age groups since individuals with more sport involvement and PA in this period have better mental health and the chance to decrease cardiovascular diseases, osteoporosis, and obesity later in life [3,4,5]. Despite the wide range of positive effects of regular physical activity, only 41% of children and adolescents in Hungary ranging from 5 to 17 years of age meet the minimum recommendations of the World Health Organization [6]. Physical education (PE) or other school-based PA would be a great opportunity to increase PA, but many schools and teachers sacrifice PE classes for other academic curricula [7].

There are other beneficial effects of PA that highlight the importance of PE and other school-based physical activities. Physical activity is related to the enhancement of cognitive function such as attention and memory, which leads to an improvement in academic performance [8]. Kibbe and colleagues [9] in their empirical investigation found that in-class PA could increase reading, math, and spelling achievements for middle school students. There is a clear relationship between grade point average (GPA) and PA, as well. Field and colleagues [10] found that higher GPA is associated with higher PA, and Lindner [11] suggested that sport participation frequency indicates better school achievements. Furthermore, studies on these topics show that there are positive relationships between PA integrated with academic subjects and children’s academic motivation [12]. There is an argument for increasing PE or other types of school-based PA without the risk of decreasing academic progress, as some studies suggest that sport involvement in PE classes might not be enough to improve academic achievement [13]. Even though there are some inconsistencies in the literature, all researchers agree that regular PA has beneficial effects on school performance besides the prevention of chronic disease and preservation of mental health. Therefore, the promotion of regular PA and sport involvement would be beneficial inside and outside of school, as well.

The role of PA in school performance is known, but the associations with learning motivation are not yet clear. Only a few studies investigate this issue. For example, PE classes focusing on learning and individual improvement were found to increase intrinsic motivation [14]. Furthermore, another study showed that PA integrated with academic subjects can positively impact children’s academic motivation [12]. We believe a deeper exploration could help elucidate the role of PA in learning motivation.

There are several motivational constructs to understand learning motivation [15]. According to Svinick and Vogler, learning motivation is goal-directed behavior which creates an interaction between learning and the environment. The researchers generally agree that highly motivated students need competence, autonomy, interest, and relatedness [15,16]. Entwistle and Kozeki [17], in an early study, conceptualized four dimensions and nine factors of motivation after conducting 1000 interviews (Table 1). According to their work, learning motivation is dependent on individual cognitive skills, social interaction, and self-awareness. Therefore, they created a three-dimension model, where the Affective Domain is described by feelings relating to the social environment (e.g., teachers, peers). The Cognitive Domain is explained by feelings relating to success, recognition, skills, and interest. The Moral Domain is described as the individual’s self-esteem, personal values, personal behavior, and self-evaluation. Entwistle and Kozeki [17] found an extra dimension as well, naming it Adult Pressure. It includes feelings that adults demand unrealistic achievements. It has a negative role in student motivation. The dimensions of learning motivation were found to be important since they have positive impacts on school performance, self-efficacy, and self-criticism [18]. On the other hand, it was also found that too much pressure from peers, teachers, and parents had a negative effect on the learning experience [18].

One of the key elements of PA and sport participation is sport motivation [19], which might influence learning motivation and school performance since students who are motivated to participate in sport or PA might exercise more and perceive the benefits of the activities that were discussed above. There are several aspects of sport motivation, but according to the Sport Commitment Model by Scanlan [20], one of the main factors to engaging in PA is sport enjoyment. Sport enjoyment is defined as “The positive affective response to a sport experience that reflects generalized feelings of joy” [20] (p. 235). The association was observed in different activity levels, from elite athletes to recreational PA participation [21]. Since sport enjoyment has a positive role in students’ PA involvement, there might be a link between enjoyment of physical activity, school performance, and learning motivation. However, to the best of our knowledge, there is only one study that investigates these associations. Souza Luján and Bossio [22] found that students who enjoyed PA had greater confidence in their skills and abilities regarding their academic performance. Thus, further analysis is needed to explore this phenomenon.

Based on the results of previous studies, we believe PA, enjoyment of physical activity, and academic performance might be associated with learning motivation. Therefore, the main goal of this study is to examine the relationships between learning motivation, physical activity, enjoyment of physical activity, and academic performance. To investigate these associations, we propose a conceptual path model (Figure 1). On this model, we hypothesize that physical activity (PA total) is associated with the four dimensions of learning motivation, and with school performance (GPA) and enjoyment of physical activity (PACES-H). Furthermore, we believe that GPA and enjoyment of physical activity are associated with learning motivation as well, and that enjoyment of physical activity is associated with GPA.

## 2. Materials and Methods

### 2.1. Participants and Procedure

The data collection occurred from September to December 2021. The online survey was voluntary, and the recruitment was via social media (e.g., Facebook) due to the COVID-19 pandemic. The schools were opened during this time and all teaching was in person. However, the schools had strict rules on outside activities, so we decided to collect our data online. The questionnaires took approximately 15–20 min to complete. All the participants were ensured that the survey was anonymous, and no personal data were collected. All the participants provided online informed consent to participate in the study. The study was conducted in accordance with the Declaration of Helsinki, and the guidelines of the Ethics Committee at University of Szeged were followed.

A total of 249 high school students completed our online survey, ranging in age from 14 to 19 years old. Respondents included 79 boys (M_age_ = 16.4; SD_age_ = 1.28) and 170 girls (M_age_ = 16.2; SD_age_ = 1.22). Moreover, 85.6% of the recruited participants came from one of the small towns in Hungary, 9.8% live in cities, and 4.5% live in the capital. Most of the students lived with their parents (71.9%), and 28.1% of participants lived with only one of their parents. The students were asked about the subjective financial status of their families: 65.6% of the sample rated themselves as middle-class, 23.9% rated themselves as upper-middle-class, 4.0% rated themselves as lower-middle-class, 5.3% rated themselves as upper-class, and 1.2% rated themselves as lower class.

### 2.2. Measures

#### 2.2.1. Sociodemographic Background and School Performance

The participants were asked about their age, gender, family status, and school performance background (“What grades do you usually get in history?”). We asked the participants about the five main subjects (mathematics, literature, history, foreign language, physical education) that all students must learn in school. Grade point average was calculated from grades of the five subjects. The Hungarian grading system includes a 1–5 scale, where 1 is the worst and 5 is the best.

#### 2.2.2. School Motivation Inventory

Learning motivation was measured by Entwistle and Kozeki’s widely used Learning Motivation Inventory [17]. The questionnaire included 60 items and ten subscales. The subscales were computed into the four dimensions of learning motivation. The Affective Domain contained Warmth, Identification, and Sociability subscales (e.g., “It is a good feeling to see my parents are happy when I do well at school”). The Cognitive Domain included subscales of Independence, Competence, and Interest (e.g., “We learn a lot of things at school that can be beneficial in real life”). The Moral Domain had subscales that included Trust, Compliance, and Responsibility (e.g., “If my peers need assistance, I always help them”). Adult Pressure was measured on only one subscale (e.g., “Adults always expect too much from their children and their students”). Each subscale included six items, and participants answered on a five-point Likert-type scale (1 = strongly disagree to 5 = strongly agree). The total score for each subscale varied between 6 and 30 and the sum of the subscales was calculated to analyze the dimensions of learning motivation. All dimension scores varied between 18 and 90, except Adult Pressure, which varied between 6 and 30. A higher score indicates a higher level of the dimension in all cases. The internal reliability in this study varied between 0.71 and 0.83.

#### 2.2.3. Physical Activity Enjoyment Scale

Enjoyment of physical activity was measured by the Hungarian version of the Physical Activity Enjoyment Scale (PACES-H). The scale was developed by Kendzierski and Decarlo [23] and modified by Motl and colleagues [24]. The second version of the scale was translated for this study. It included 16 items answered using a five-point Likert-type scale. The answer categories ranged from 1 (strongly disagree) to 5 (strongly agree). In the validation process, we followed the general process of scale translation [25]. The questionnaire is a unidimensional scale, and the total score could vary from 16 to 80. A higher score indicated a higher level of enjoyment of physical activity. The scale showed an excellent model fit on confirmatory factor analysis (x^2^(80) = 91.0, *p* = 0.18; CMIN/d.f. = 1.12; CFI = 0.99; TLI = 0.99; SRMR = 0.03; RMSEA = 0.02) and high internal reliability (Cronbach alpha = 0.93; average variance extracted = 0.53; construct reliability = 0.92).

#### 2.2.4. International Physical Activity Questionnaire

The International Physical Activity Questionnaire (IPAQ-SF) was used to assess PA. The short form of the questionnaire was validated earlier in the Hungarian population [26]. The questionnaire consisted of seven items, and it measured the frequency (day/week) and time (hour and minutes/day) over the last seven days of walking and moderate- and vigorous-intensity physical activity (e.g., “How many days did you perform vigorous physical activity that made you feel tired or breathless during the past week?” [27] (p. 2)). The last item of the IPAQ-SF measures sedentary behavior, but it was not involved in this study and is not part of the summary score. The metabolic equivalent of a task (MET) was used to estimate the energy expenditure of physical activity. Following the official International Physical Activity Questionnaire protocol (www.ipaq.ki.se, accessed on 21 January 2022), MET min/week values were calculated for walking and moderate- and vigorous-intensity exercise. Then, we summed up the three values and obtained the total number of physical activities for each participant. Intraclass correlation coefficient (ICC) was used as a reliability measure of IPAQ-SF. The total PA of this study showed a moderate reliability (ICC = 0.52), which is consistent with previous studies [26,28].

### 2.3. Statistical Analysis

After data collection and coding, we used Jamovi 2.0 for Mac to analyze our data. Descriptive statistics were used to understand the characteristics of our data. Correlation analysis was used to see the relationships between variables. After primary analysis, we tested our hypothesized model using path analysis. Widely used fit indexes were calculated to interpret the results, including chi-square (x^2^), relative chi-square divided by the degree of freedom (CMIN/d.f), root mean square error of approximation (RMSEA), a non-normed fit index called the Tucker–Lewis index (TLI), comparative fit index (CFI), and standardized root mean square residual (SRMR). The acceptable range of the fit indexes was as follows: x^2^ was acceptable if nonsignificant, but we should acknowledge that it is very sensitive to the sample size; hence, CMIN/d.f. was also used [29]. If the CMIN/d.f. value is lower than 3, then our model is acceptable. RMSEA values are acceptable if lower than 0.08 [30]. The Tucker–Lewis Index is considered acceptable above 0.90, but a good fit should be considered above 0.95 [31]. The acceptable value for CFI is above 0.90. If the SRMR value is less than 0.08, we can accept the model fit [32].

## 3. Results

### 3.1. Descriptive Statistics and Gender Differences

Table 2 displays the mean, standard deviation, minimum, maximum, skewness, and kurtosis of the four dimensions of school motivation, PACES-H, GPA, and total values of PA measured in MET min/week. The participants of this study reported moderate levels of Affective Domain (M = 64.08), Cognitive Domain (M = 57.03), Moral Domain (68.50), and Adult Pressure (M = 16.95). The sample perceived a relatively high level of enjoyment of physical activity (M = 61.21), and they were considered to have good GPAs (M = 4.43). The participants’ average PA total was 2743.47 MET min/week. Our skewness and kurtosis values were acceptable (−1.5, 1.5) for normal distributions [33].

### 3.2. Bivariate Correlations

Correlation analysis examined the relationship between learning motivation, PA, enjoyment of physical activity, and GPA (Table 3). A positive and significant association was found between Affective Domain, Cognitive Domain, Moral Domain, PACES-H, and GPA. Adult Pressure was negatively associated with Affective Domain, Cognitive Domain, Moral Domain, GPA, and PACES-H. Physical activity did not correlate with the dimensions of learning motivation, but it showed a significant relationship with GPA and PACES-H. Of all the significant relationships, the strongest correlation was found between the dimensions of learning motivation.

### 3.3. Path Analysis

Path analysis was used to determine which variables play a role in learning motivation. The hypothesized model did not adequately fit to our data (x^2^(1) = 5.41, *p* = 0.01; CMIN/d.f. = 5.41; CFI = 0.98; TLI = 0.7; SRMR = 0.03; RMSEA = 0.15). Therefore, the direct path between PA total and learning motivation was discarded since these variables did not correlate. Our final model showed an acceptable model fit (x^2^(4) = 8.54 *p* = 0.07; CMIN/d.f. = 2.13; CFI = 0.99; TLI = 0.96; SRMR = 0.03; RMSEA = 0.05). As expected, GPA and enjoyment of physical activity positively influenced the Affective Domain, Cognitive Domain, and Moral Domain (Figure 2). Furthermore, PACES-H had a negative relationship with Adult Pressure. However, GPA did not have a significant relationship with Adult Pressure. PA total was found to be a significant predictor of GPA and PACES-H. Our model accounted for 15% of the variance in Affective Domain, 15% in Cognitive Domain, 15% in Moral Domain, and 5% in Adult Pressure.

## 4. Discussion

The goal of this study was to understand the roles of PA, enjoyment of physical activity, and GPA in the dimensions of learning motivation [17]. The hypothesized model was tested via path analysis, and the results indicate that enjoyment of physical activity and GPA are associated with learning motivation. However, PA only indirectly supports learning motivation, since there was no correlation with the learning motives, but it was positively correlated with the enjoyment of physical activity and GPA.

The positive effects of physical activity were discussed earlier, and its role in the prevention of chronic diseases, preserving mental well-being, and nurturing better academic performance is well established [1]. Some studies also suggested that PA has a role in learning motivation, as well [13]. However, our results did not show any relationship between PA and the dimensions of learning motivation. We assume that the reason behind this result is that individuals in this study might not participate in any other sport or PA outside of school. According to Bunketorp-Kall [13], school-based PA alone might not be enough to improve the motivation for learning. Even though there was no correlation with the dimensions of learning motivation, PA had a positive influence on GPA and enjoyment of physical activity. Thus, it is indicated that a higher level of PA increases school grades and the enjoyment of physical activity. Previous studies highlighted the positive influence of physical activity on academic performance. These studies are in line with our results since they found that individuals with a higher level of PA have higher GPA and stronger school performance than those who have a lower level of PA [8,9,10]. Several motivational studies showed the importance of enjoyment of sports and PA. These studies also found that individuals with higher levels of PA have higher levels of physical activity enjoyment [9,34,35].

We must point out an important finding of our study. There is an indirect relationship between PA and the dimensions of learning motivation. Physical activity increased GPA and enjoyment of physical activity, and both variables were associated with learning motivation. Thus, it seems as though individuals with higher PA levels have better school performance and perceive greater enjoyment of physical activity, which could increase the cognitive affective and moral dimensions of learning motivation.

The relationship between GPA and Cognitive Domain, Affective Domain, and Moral Domain was not surprising, since previous studies showed the importance of motivation in school performance [36,37]. However, we must acknowledge that GPA and Adult Pressure had a negative but nonsignificant relationship in this study. Adult Pressure was characterized by unrealistic pressure and constraints from adults, and it seems that students with better grades perceive less pressure from their parents and teachers, but future studies should investigate these relationships in depth.

This was the first study to investigate the association between learning motivation and the enjoyment of physical activity using PACES-H. There were positive associations with school performance and cognitive, affective, and moral dimensions of motivation. Individuals with higher levels of physical activity enjoyment had better grades and perceived higher levels of empathy and interest. They had more ideas and drive to develop new skills and felt more responsible for their actions. Negative associations were found between Adult Pressure and enjoyment of physical activity. We believe the negative relationship indicates that enjoyment of physical activity might have a preventive role concerning Adult Pressure. Individuals who enjoy sports seem to perceive less pressure from their parents and teachers. The reason that such enjoyment might have a preventive role regarding Adult Pressure is that these individuals perceive more intrinsic motivation than those who do not enjoy sports and, in this case, than the students participating sports because of external reasons (e.g., pressure from their parents). Previous studies also supported these results [20,38]. Based on this finding, we must highlight the importance of intrinsic motivation, as it not only increases sport motivation, but it also has a more complex role. It could increase levels of learning motivation and academic performance and decrease pressure from peers among students.

Our study has limitations that need addressing. First, the generalizability is limited because of the convenience sampling, and because girls are primarily represented in this study. Second, the methods of data collection utilizing social media platforms may have narrowed down participation. We must acknowledge that the data collection occurred during the COVID-19 pandemic, which might influence our results, but our study design did not involve the pandemic as a covariate. Thus, further analysis should address this issue. Another limitation is the relatively small effect size of the path analysis. Finally, we must acknowledge that the self-administered questionnaire is a limitation as well, since it fails to give a realistic picture of the feelings of the students. To remove limitations, our study will continue in the future. Our goal is to increase the sample size and use objective PA measures (e.g., accelerometer) to develop a deeper understanding of the associations between physical activity, learning motivation, and learning strategies.

## 5. Conclusions

In summary, PA was not directly associated with learning motivation as we hypothesized, but it had a positive effect on GPA and enjoyment of physical activity, increasing cognitive, affective, and moral dimensions of motivation, which indicates an indirect relationship with learning motivation. Another finding of this study was the role of enjoyment of physical activity. It turned out to have a preventive role regarding pressure from adults since it strengthens the intrinsic motivation of the students.

We believe our study provides useful information for researchers, teachers, and any other professionals who can encourage adolescent children to participate in regular physical activity and sports, since the benefits of such participation could positively affect students’ engagement and achievements in learning.

## Figures and Tables

**Figure 1 children-09-00320-f001:**
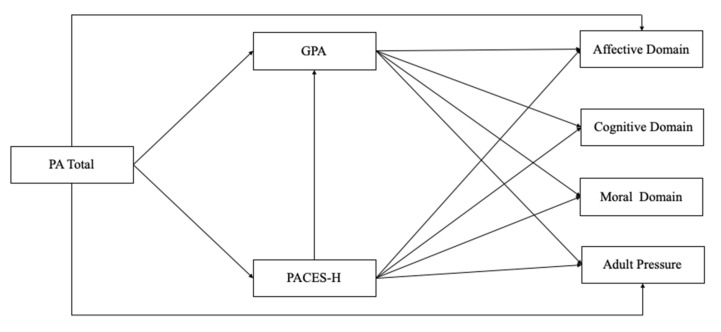
Hypothesized path model and its associations between PA, PACES-H, GPA, and the dimensions of learning motivation. note. PA = Physical Activity; PACES-H = Physical Activity Enjoyment Scale; GPA = Grade Point Avarage.

**Figure 2 children-09-00320-f002:**
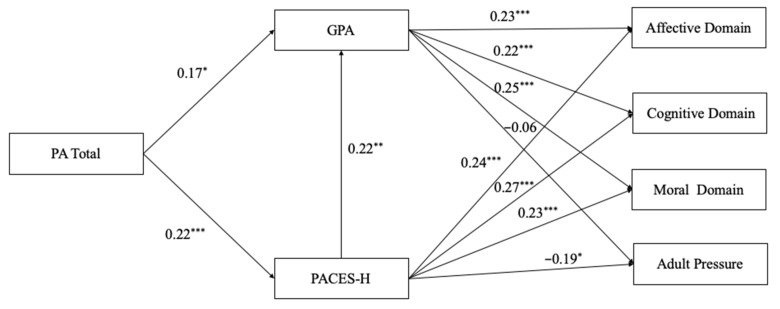
Results of path analysis between learning motivation, PACES-H, GPA, and PA total. Note. * *p* < 0.05; ** *p* < 0.01; *** *p* < 0.001.

**Table 1 children-09-00320-t001:** Dimensions of learning motivation by Entwistle and Kozeki, reprinted with permission from Ref [17] (p. 244). Elsevier, 2022.

Dimensions	Description of the Sources of Motivation
Affective Domain	
Warmth	Encouragement and interest of parents
Identification	Feeling empathy with teachers and wanting to please them
Sociability	Enjoying collaborative work and activities with peers
Cognitive domain	
Independence	Satisfaction from working things out without help from others
Competence	Rewards from recognition of developing knowledge and skills
Interest	Enjoyment derived from ideas
Moral Domain	
Trust	Satisfaction from accomplishing things thoroughly and well
Compliance	Preferring the security of behaving according to defined rules or norms
Responsibility	Accepting the consequences of actions and monitoring own behavior
Adult Pressure	Feeling that the adults demand unrealistic achievements

**Table 2 children-09-00320-t002:** Descriptive statistics of the study variables.

	Affective Domain	Cognitive Domain	Moral Domain	Adult Pressure	PACES-H	GPA	PA Total (MET min/week)
Mean	64.08	57.03	68.50	16.95	61.21	4.43	2743.47
SD	10.07	10.05	10.09	4,98	12.82	0.49	2559.00
Minimum	36.00	28.00	28.00	6.00	23.00	2.40	15.30
Maximum	88.00	82.00	89.00	30.00	80.00	5.00	9607.00
Skewness	−0.16	0.06	−0.38	0.14	−0.57	−0.98	1.25
Kurtosis	−0.24	−0.21	0.23	−0.47	−0.17	0.92	1.41

**Table 3 children-09-00320-t003:** Correlation of learning motivation, PACES-H, GPA, and physical activity.

	1	2	3	4	5	6
1. Affective Domain	-					
2. Cognitive Domain	0.58 ***	-				
3. Moral Domain	0.66 ***	0.58 ***	-			
4. Adult Pressure	−0.56 ***	−0.48 ***	−0.47 ***	-		
5. PACES-H	0.34 ***	0.35 ***	0.31 ***	−0.25 ***	-	
6. GPA	0.25 ***	0.29 ***	0.30 ***	−0.15 *	0.21 **	-
7. PA Total	0.05	0.05	0.08	0.00	0.26 ***	0.20 **

Note. * *p* < 0.05; ** *p* < 0.01; *** *p* < 0.001.

## Data Availability

Not applicable.

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
