# Peer review of "The Role of Physical Activity, Enjoyment of Physical Activity, and School Performance in Learning Motivation among High School Students in Hungary"

_children, 2022, doi:10.3390/children9030320_

Round 1

Reviewer 1 Report

Title: The role of physical activity, enjoyment of physical activity, school performance on learning motivation among high-school students in Hungary

The manuscript entitled, The role of physical activity, enjoyment of physical activity, school performance on learning motivation among high-school students in Hungary, presents findings from a cross-sectional study. Study recruitment occurred through Facebook with several questionnaire administered to assess current physical activity participation, enjoyment of physical activity, school performance and learning motivation. The study design notably, recruitment and data collection via social media, is a limitation and warrants a bit more explanation. The study findings obtained however, add somewhat useful information to the literature.

Comments are provided below:

Title: Should the word “and” be added in the title so that it reads, “The role of physical activity, enjoyment of physical activity, and school performance on learning motivation among high-school students in Hungary”?

Page 1, lines 28-29, “early ages” is mentioned yet the study is focused on high school age students. Perhaps background information pertaining to older children and adolescents might be a bit more relevant to include as “early ages” may be interpreted as young as preschool.

Page 1, lines 31-33: “…41% of the children…”; it may be helpful to know the age range this is referring to.

Methods and Results

A total of 5 min to complete all questionnaires seems inaccurate considering one of the three questionnaires administered consists of 60 questions.

It would be informative to know how this response rate represents the high school students in the area (i.e., 249 respondents out of ~3,000 high school students located in the capital of Hungary).

Why did recruitment occur through Facebook rather than schools?

Page 5, line 168: Please consider changing “we’re” to “we”

Page 6, line 212-213: “physical enjoyment of activity” should likely read “enjoyment of physical activity”

Discussion

Page 7, line 231: Please consider changing “seen” to “presented” or “discussed”

Page 7, lines 249-251, 262-263: Please revise the wording of this sentence

Page 7, lines 259 and 264: “perceives” should be “perceive”

Page 7, line 261: Please consider changing “It” to “This”

Page 7, line 270: Please consider revising “The reasoning might enjoyment has a preventive role” to “The reasoning enjoyment might have a preventive role”

Page 8, line 277: Please consider revising “what” to “that”

Data collection occurred from September to December 2021. I am wondering how COVID may have played a role in some of the responses. What were the circumstances regarding schools and school-based activities during this time (i.e., all teaching in-person, hybrid, resumption of sports and after-school activities)? At the minimum, a brief explanation of what the situation was in terms of COVID in this region of Hungary/schools should be mentioned and potential for how it may have impacted student responses.

The abbreviation of physical activity (PA) is inconsistent throughout the paper. The same observation applies to physical education (PE) and grade point average (GPA). Terms should be spelled out and abbreviated within parentheses at the first mention and then abbreviated from there on out (except for when a sentence begins with the abbreviated terms).

Author Response

Reviewer 1.

First, I would like to say thank you for your comments and suggestions. All of them have been taken into account and it was very helpful to improve the quality of the article. All changes were marked red in the manuscript.

Title: Should the word “and” be added in the title so that it reads, “The role of physical activity, enjoyment of physical activity, and school performance on learning motivation among high-school students in Hungary”?

Thank you for your suggestion "and" was added to the title 

Page 1, lines 28-29, “early ages” is mentioned yet the study is focused on high school age students. Perhaps background information pertaining to older children and adolescents might be a bit more relevant to include as “early ages” may be interpreted as young as preschool.

Thank you for your comment on this issue. The study is about adolescents, but it is important to start PA in early ages as preschool kids. We make this statement clear on line 30.   

Page 1, lines 31-33: “…41% of the children…”; it may be helpful to know the age range this is referring to.

Thank you for your suggestion "children and adolescents ranging from 5-17 years of age " was added to the text.

A total of 5 min to complete all questionnaires seems inaccurate considering one of the three questionnaires administered consists of 60 questions.

Thank you for your comment. The five minutes was changed to 15-20 minutes which is reflects an accurate time to complete this questionnaire.

It would be informative to know how this response rate represents the high school students in the area (i.e., 249 respondents out of ~3,000 high school students located in the capital of Hungary).

Thank you for your comments. We did not include a response rate to this study, since we only have an approximate number. For example, there are 2243 high schools in Hungary, and it includes approximately 411.000 students. According to this result, the response rate would be 0,06, which is so small that we can't add relevant information. However, we added to the limitation that, the generalizability is limited because of the convenience sampling.

Why did recruitment occur through Facebook rather than schools?

Thank you for the reviewer’s comment on this issue. All schools were opened during our data collection, but the 4th wave of the pandemic was on therefore we decided it is easier to collect data online.

Page 5, line 168: Please consider changing “we’re” to “we”

Page 6, line 212-213: “physical enjoyment of activity” should likely read “enjoyment of physical activity”

Page 7, line 231: Please consider changing “seen” to “presented” or “discussed”

Page 7, lines 249-251, 262-263: Please revise the wording of this sentence

Page 7, lines 259 and 264: “perceives” should be “perceive”

Page 7, line 261: Please consider changing “It” to “This”

Page 7, line 270: Please consider revising “The reasoning might enjoyment has a preventive role” to “The reasoning enjoyment might have a preventive role”

Page 8, line 277: Please consider revising “what” to “that”

Thank you for your deep analysis. All the lines were corrected.

Data collection occurred from September to December 2021. I am wondering how COVID may have played a role in some of the responses. What were the circumstances regarding schools and school-based activities during this time (i.e., all teaching in-person, hybrid, resumption of sports and after-school activities)? At the minimum, a brief explanation of what the situation was in terms of COVID in this region of Hungary/schools should be mentioned and potential for how it may have impacted student responses.

Thank you for raising this issue. We're aware that the COVID pandemic might be influenced are results, but our study design did not involve covid as a predictor. However, we make it clear in the Limitation ("We must acknowledge that the data collection occurred during the COVID-19 pandemic, which might influence our results, but our study design did not involve the pandemic as a covariate. Thus, further analysis should address this issue. ") and on the procedure as well (via COVID pandemic. The schools were opened during this time, and all teaching was in person. However, the schools had strict rules of outside activities and invented quests, hence we decided to collect our data online.).

The abbreviation of physical activity (PA) is inconsistent throughout the paper. The same observation applies to physical education (PE) and grade point average (GPA). Terms should be spelled out and abbreviated within parentheses at the first mention and then abbreviated from there on out (except for when a sentence begins with the abbreviated terms).

Thank you for your comments, all abbreviation was corrected.

Reviewer 2 Report

The paper describes the relationship between physical activity, school performance and learning motivation. The authors have collected data from a reasonable sample of young people which is good but the rationale for the direction of effects examined needs to be more fully explained. I have some specific suggestions/comments:

Abstract needs a sentence of implications – what contribution do the findings make?

I am not convinced as to the logic of the RQ. Why would PA predict learning motivation? This is not well explained in the paper and needs to be revised to make the rationale clearer.

Learning motivation needs to be more clearly defined/explained. And why enjoyment of PA would have a direct path to GPA needs a fuller explanation.

Line 125 – it is not clear what “to the easier analysis” means when describing GPA. Please also include more detail so it is clear what the min/max/scoring means for this variable (e.g. does an A get a score of 5?).

It would be useful to include an example item for the Learning motivation inventory and also specify if a high score was positive or not.

Please include reliability of the IPAQ. It might also be useful to present the data as mins per day, rather than per week as that aligns with the PA guidelines so is more translatable.

Line 168 should be we, not we’re

I think the notation for chi-square is incorrect. This should show as χ2

Line 212 – I think should be enjoyment of physical activity.

The results are mostly well presented and clear. I wonder if the authors tested the model where learning motivation predicted GPA? This seems to be more intuitive and aligns with the evidence in the discussion where motivation is suggested as important for school performance.

While there are a number of statistically significant paths, the magnitude of relationships is small. Please include this in your discussion/interpretation of the findings.

It might also be worth discussing the use of objective measures of PA in future research.

It would also be good to discuss theoretical models for the relationship in the discussion.

Author Response

Reviewer 2.

First, I would like to say thank you for your comments and suggestions. All of them have been taken into account and it was very helpful to improve the quality of the article. All changes were marked red in the manuscript.

Abstract needs a sentence of implications – what contribution do the findings make?

Thank you for your suggestion. We added an extra sentence of implications: "We believe our findings highlight the benefits of physical activity and enjoyment of physical activity in learning environments"

I am not convinced as to the logic of the RQ. Why would PA predict learning motivation? This is not well explained in the paper and needs to be revised to make the rationale clearer.

Thank you for raising this issue. We added more articles on this topic, to make the associations on PA and learning motivation clear. However, we would like to highlight that there are only a few studies to investigate this issue.

Learning motivation needs to be more clearly defined/explained. And why enjoyment of PA would have a direct path to GPA needs a fuller explanation.

Thank you for your comment we added more explanation to the text ("According to Svinick and Vogler learning motivation is a goal-directed behavior, which creates an interaction between the learning and the environment. The researchers generally agreed that highly motivated students need competence, autonomy, interest, and relatedness").

            Best of our knowledge there is only one study that investigated the enjoyment of PA and school performance. The direct path between the PA and GPA is a hypothesis that comes from the phenomenon that higher motivation for sport increases sport participation and that clearly increases school performance as well. We make this association clear in the text (Lines 82-84). 

Line 125 – it is not clear what “to the easier analysis” means when describing GPA. Please also include more detail so it is clear what the min/max/scoring means for this variable (e.g. does an A get a score of 5?).

Thank you for your comment. We added more information on this topic: The Hungarian grade system includes a 1-5 scale 1 is the worst and 5 the best.

It would be useful to include an example item for the Learning motivation inventory and specify if a high score was positive or not.

Thank you for your comments on this issue. We added one example to each dimension of learning motivation. Furthermore, we expanded the description of the school motivation Inventory, by adding the scaling of the subscales and dimensions.  "The total score for each subscale was varied between 6 and 30 and the sum of the subscales was calculated to analyze the dimension of learning motivation. All dimension scores varied between 18 and 90, except Adult Pressure, which varied between 6 and 30. A higher score indicates a higher level of the dimension in all cases."

Please include the reliability of the IPAQ. It might also be useful to present the data as mins per day, rather than per week as that aligns with the PA guidelines so is more translatable.

Thank you for your suggestions. We measured each type of physical activity with the frequency with day per week and time per day (hours and minutes/day), then we calculated the MET score following the IPAQ guidelines. We believe in this case we get a full picture of the participant's PA, however, in the future analysis, we will consider your suggestions. Intraclass correlations were added as reliabilities to IPAQ. "Intraclass correlation coefficient (ICC) was used as a reliability measure of IPAQ-SF. The total PA of this study showed moderate reliability (ICC=0.52), which is consistent with previous studies"

Line 168 should be we, not we’re

Thank you for your deep analysis. The issue was corrected

I think the notation for chi-square is incorrect. This should show as χ2

Thank you for your comment. The issue was corrected

Line 212 – I think should be enjoyment of physical activity.

Thank you for your comment. The issue was corrected

The results are mostly well presented and clear. I wonder if the authors tested the model where learning motivation predicted GPA? This seems to be more intuitive and aligns with the evidence in the discussion where motivation is suggested as important for school performance.

Thank you for your comment. In this research we decided to GPA should predict learning motivation, since other variables were also involved in this study. In future research, we will create other paths to see the relationship between these variables in both ways.

While there are a number of statistically significant paths, the magnitude of relationships is small. Please include this in your discussion/interpretation of the findings.

Thank you for your comment on this issue. We added the small magnitude as a limitation.

It might also be worth discussing the use of objective measures of PA in future research.

Thank you for your comment. I believe using objective PA measures (accelerometer) would be ideal for future studies. Thank you, we added as a future direction